# Dredging activity and associated sound have negligible effects on adult Atlantic sturgeon migration to spawning habitat in a large coastal river

Matthew Balazik[1,2]*, Michael Barber[2], Safra Altman[1], Kevin Reine[1], Alan Katzenmeyer[1], Aaron Bunch[3], Greg Garman[2]

1 Engineer Research and Development Center, United States Army Corps of Engineers, Vicksburg, MS, United States of America, 2 Rice Rivers Center, Virginia Commonwealth University, Charles City, VA, United States of America, 3 Virginia Department of Game and Inland Fisheries, Charles City, VA, United States of America

* Matthew.T.Balazik@usace.army.mil

**Data Availability Statement:** All relevant data are within the manuscript and its Supporting Information files.

## Abstract

Dredging is considered a major threat/impedance to anadromous fish migrating to spawning habitat. Due to the perceived threat caused by dredging, environmental windows that restrict dredge operations are enforced within many rivers along the east coast. However, it is generally unknown how anadromous fish react to encountering an active dredge during spawning migrations. Atlantic sturgeon (ATS) are an endangered, anadromous species along the Atlantic slope of North America. To determine if and how an active dredge may affect ATS spawning migration, a Vemco Positioning System array was deployed around an active hydraulic-cutterhead dredge that adult ATS must traverse to reach spawning habitat in the James River, VA. Telemetry data showed that all ATS that entered the study area survived. ATS that migrated upstream during dredge operations (N = 103) traversed the dredge area and continued upstream to spawning habitat. Many ATS made multiple trips through the study area during dredge operations. There was no noticeable difference in swim behavior regardless of whether the dredge was absent or working within the study area. We suggest that dredging in the lower James River does not create a barrier for adult ATS migrating to spawning habitat or cause adults to significantly modify swim behavior. This is the first study to utilize fine-scale telemetry data to describe how an organism moves in relation to an active dredge. This methodology could be used to describe dredge-sturgeon interactions on different life stages and in other locations and could be expanded to other aquatic organisms of concern.

## Introduction

Through Coastal Zone Management Acts, many states have implemented time-of-year restriction windows, or environmental windows, on dredging projects to minimize possible negative

**Funding:** This study was funded by: grant #NA13NMF4720037 National Oceanic and Atmospheric Administration, Virginia Department of Game and Inland Fisheries to MTB and GCG and grant #DOERFY17-10 Engineer Research and Develop Center to MTB and SF. The funders provided a role in the preparation of the manuscript. The funders had no additional role in study design, data collection and analysis, or decision to publish.

**Competing interests:** The authors declare that they have no conflict of interest.

impacts they may have on natural resources. In 2012, the National Marine Fisheries Service declared that distinct population segments of Atlantic sturgeon *Acipenser oxyrinchus oxyrinchus* (ATS) were either threatened or endangered throughout their entire range along the Atlantic coast [1–2]. ATS are both anadromous and philopatric species meaning they return to upstream freshwater reaches in their natal streams to spawn [3]. Currently, areas along the U. S. east coast have established riverine environmental windows for the spring anadromous fish spawning run. When anadromous fish windows were implemented it was thought that ATS spawning was limited to the spring season. However, studies have shown that in some river systems, ATS spawning occurs during both the spring and fall seasons [4–9]. There are no riverine environmental windows currently in place to protect fall spawning ATS migrating to spawning habitat. Although the supporting data are sparse, dredging is listed as a serious threat to ATS recovery [3] and resource managers are concerned about potential impacts an active dredging operation may have on fall run adult ATS migrating to spawning habitat.

Hydraulic-cutterhead dredging has occurred in the James River during the summer-fall every year since 1998 except for 2000, 2006 and 2013. Dredging occurs annually between river kilometer (rkm) 60 and 80 of the James River, Virginia, due to shoaling (i.e., buildup of bottom sediments in river channels) in the federal navigation channel. Dredging typically begins soon after the spring anadromous fish environmental window ends in July and continues through November to create safe passage for shipping. Fall spawning ATS usually stage in the lower part of the James River near rkm 30 until water temperatures decrease below 28˚C during August-September and then move upstream above rkm 130 to spawn [7,10]. This creates a situation where adult ATS must traverse annual dredging operations to reach spawning habitat. Since 2016, collections of fall spawn young-of-year and age-1 ATS were verified using genetics and show successful spawning during the past few years when dredging occurred during the fall spawning migration (Matthew Balazik, Unpublished Data). Even though successful reproduction by fall ATS exposed to dredging operations has been proven, information is needed to elucidate potential impacts of dredging in a riverine environment on migrating ATS. The goal of this study was to determine how migrating adult ATS move in relation to an active hydraulic-cutterhead dredge by utilizing Vemco Positioning System (VPS) technology. Implications of this work will help determine if environmental windows are an effective management policy.

## Methods

This work was carried out in guidelines set by Virginia Commonwealth University's Institutional Animal Care and Use Committee (#AD20127) and the National Marine Fisheries Service endangered species permit (#16547).

### Study area

The study area is a tidal, 12 km reach of the James River federal navigation channel (Figs 1 and 2). Federal regulations state that a 91.4 m wide, 7.6 m deep channel must be maintained for navigation. Due to severe shoaling in the federal navigation channel, dredging occurs annually from July through November to maintain legally required channel depths. In 2017 two portions of the 12 km stretch were dredged (Fig 2). The average river width of the 12 km stretch is 4.3 km. Because the study area was located in a public area, no special permits were required to access the field site.

### Vemco Positioning System array

A VPS utilizes trilateralization technology to estimate down to 1 m spatial resolution of telemetered objects within a study area [11]. This technique requires acoustic receivers to be

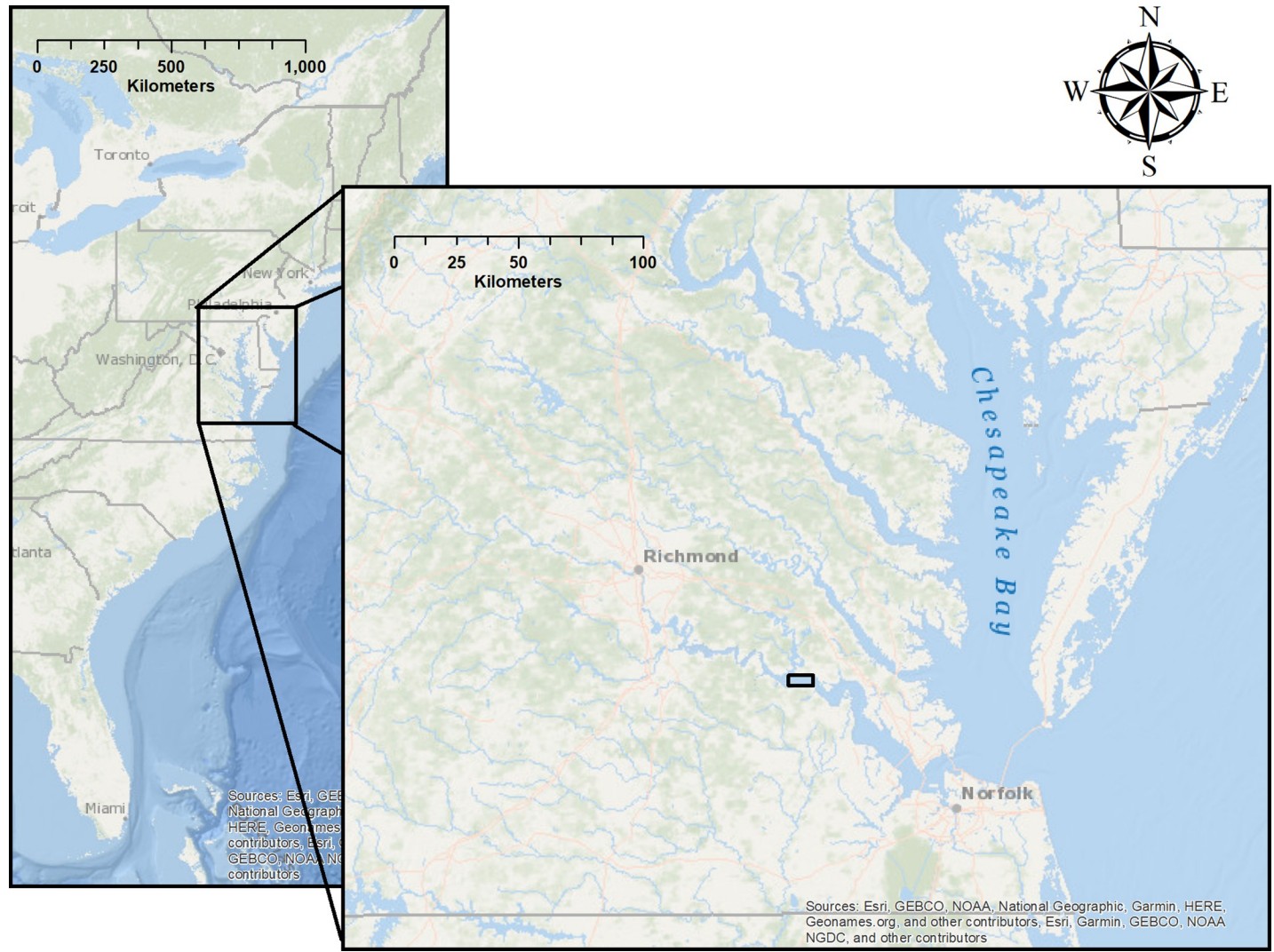

**Fig 1. Map showing study location.** Basemap reprinted from Ocean Basemap under a CC BY license, with permission from ESRI, original copyright 2012.

strategically placed to allow a single ping from a telemetry tag to be detected by at least three receivers. The position of the tag can be estimated using difference in arrival time to multiple receivers. The tagged organisms in this study were verified to be sexually mature fall spawning ATS when the telemetry tag was placed. During previous years, Virginia Commonwealth University has surgically implanted Vemco telemetry tags into various sized ATS in the James River [5,10]. There were about 140 adult James River population ATS at large of which about 100 enter the James River annually to spawn at the time of this project.

There were not enough receivers to cover the entire dredge area (Fig 2). The VPS was deployed around the lower dredge area where the river was narrower and would provide coverage for almost the entire width of the river. A 37 bottom mounted VPS receiver array with collocated synchronization tags was deployed around rkm 65 to monitor movements of telemetered ATS in the lower dredge area (Fig 2). One reference tag was placed in the middle of the array to improve positioning data of the ATS (Fig 2). The lower dredge area was chosen because the width of the river is narrower; therefore, the VPS would cover more of the overall

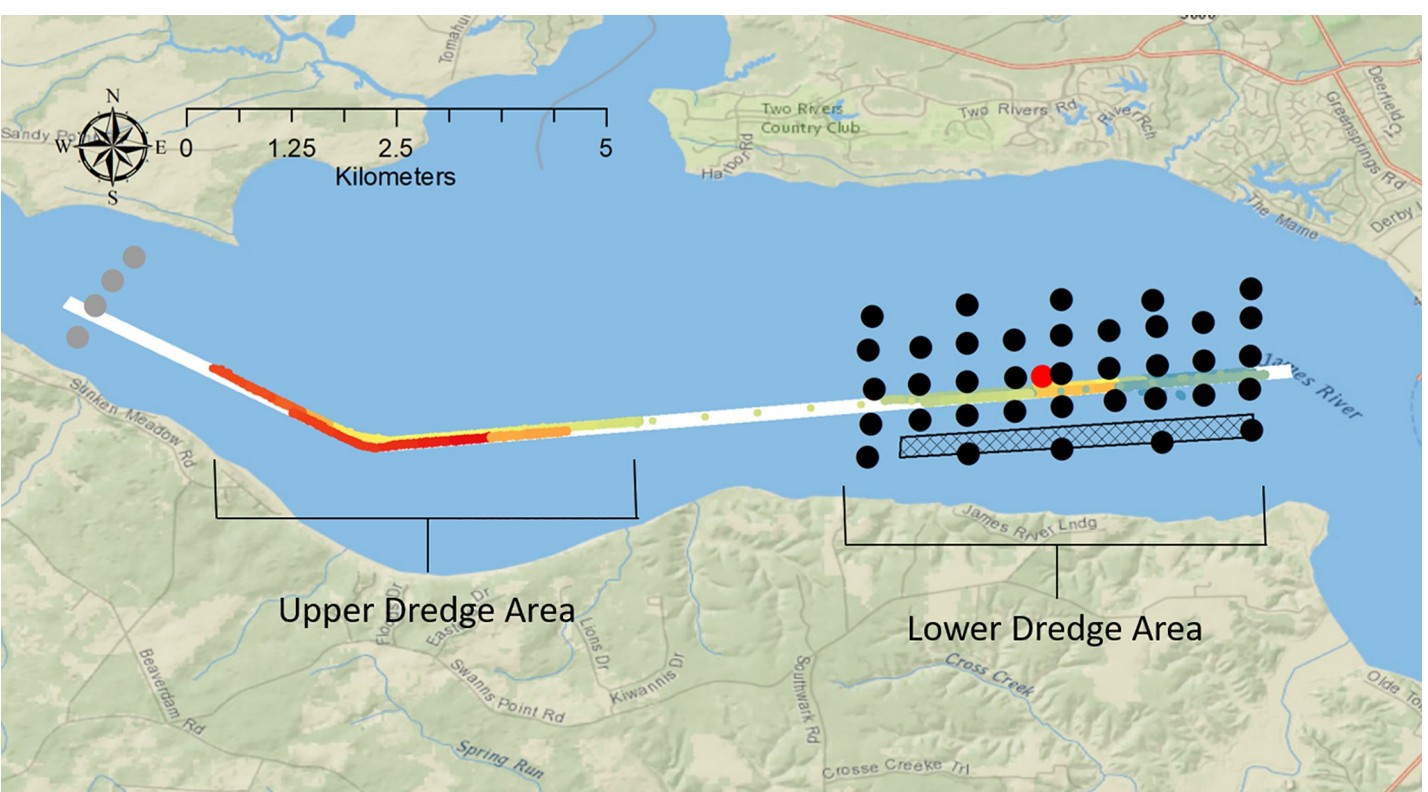

**Fig 2. Map of study design.** The white polygon is the navigation channel and the colored dots within the channel are where the hydraulic-cutterhead dredge was working during the study period. The colored dots within the channel show where the dredge was working. The different colors show where the dredge was working each day. The black dots are collocated receiver locations of the Vemco Positioning System array. The red dot is a reference tag to assist with determining fine-scale positions for the telemetered Atlantic sturgeon. The gray dots are receiver locations for the supplemental array upstream of the dredge area. The black grid in the southern area of the array is where the dredged material from the lower dredge area was placed. Basemap reprinted from Ocean Basemap under a CC BY license, with permission from ESRI, original copyright 2012.

width of the river. It is very important to minimize receiver movements when conducting VPS studies [11]. VPS Receivers were placed inside 0.7–1 m long, 8 cm diameter PVC pipes with the receiver hydrophone exposed on one end with the other end of the pipe embedded in a 61X61X12 cm concrete slab weighing about 45 kg (Fig 3). Rebar cross members that extend about 50 cm from each direction of the concrete slab were added to strengthen the concrete. The reinforced concrete slab and extended rebar helped prevent the receiver from moving or tipping over in the water current. Rebar bent in a U-shape was placed in the concrete to provide handles to make moving the receiver stands easier and create a grappling point. A 30 m piece of positively buoyant rope was secured to the receiver stand and the other end of the rope was secured to a 2 kg anchor or cinder block (Fig 3). The gear was retrieved by grappling the rope between the two anchors. The VPS array deployment was initiated on July 20, completed on August 6 and was removed October 29, 2017.

A supplemental receiver gate comprising 4 receivers was placed on July 20 about 2 km upstream of the upper dredge area to provide additional receiver coverage at the upper dredge area. (Fig 2). Each supplemental receiver was affixed to the bottom with an 11 kg anchor. The supplemental gate was used to determine when ATS passed through the upper dredge area but did not provide fine-scale positions.

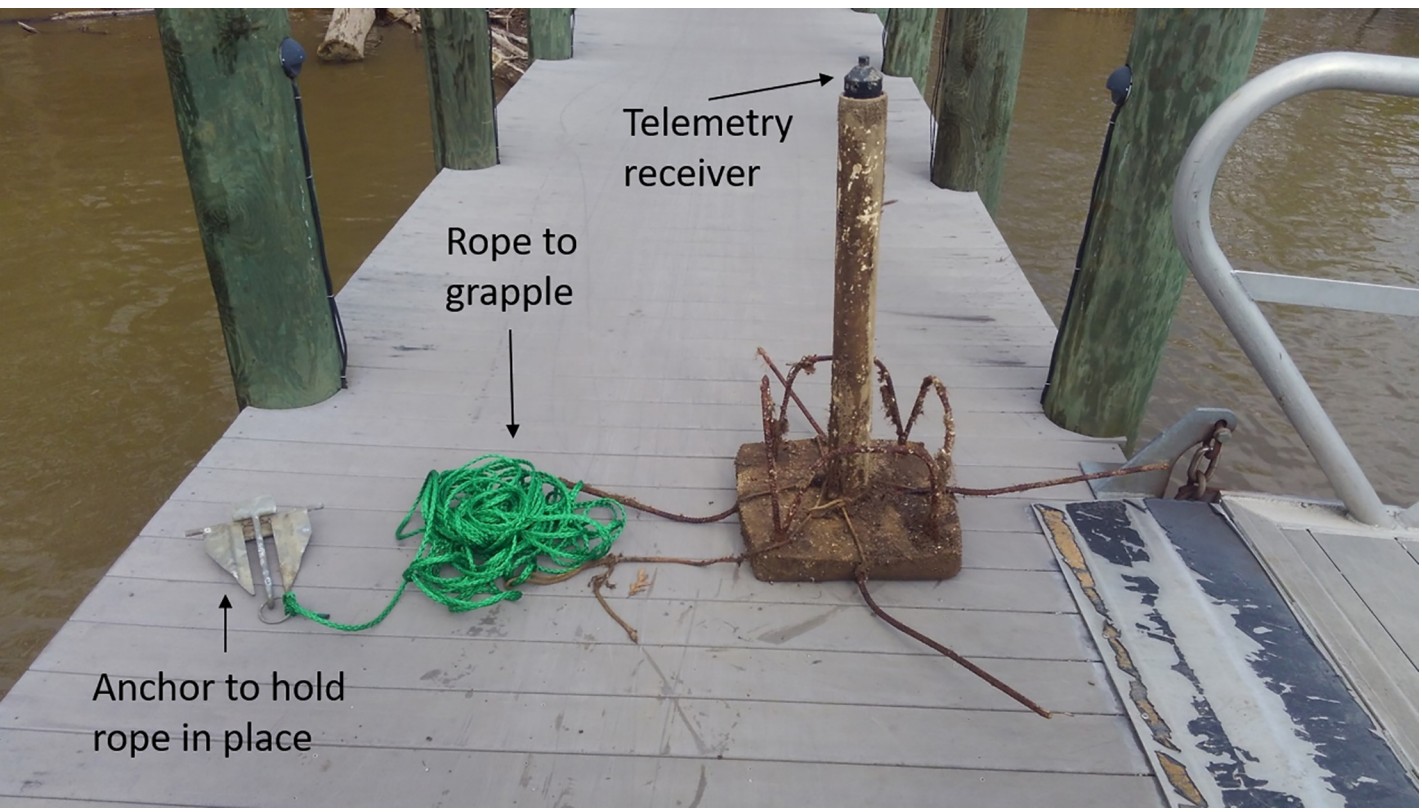

**Fig 3. Setup to reduce movements of receives in the Vemco Positioning System array.**

### Position analysis

Other telemetry studies have been used to describe sturgeon swim behavior [12,13] but none have focused on movements around something that could be perceived as a threat such as a dredge. VPS positions used for data analysis were vetted by only using positions estimated to be within the array grid (Fig 4). Positions were vetted further following methods by Roy et al. [11] using horizontal positioning error scores. The Optimized Hot Spot Analysis tool (ESRI ArcMap 10.5.1 Pro) was used to determine areas that ATS more frequently used while moving through the study area. The Optimized Hot Spot Analysis tool removes the subjective aspect of determining point clusters by assigning confidence values to each cell as either a significant hot spot, a non-significant spot, or a significant cold spot. This technique utilizes inferential statistics that applies $z$ and $p$ scores to hexagon polygons using Getis-Ord Gi* Cluster Analysis [14]. The hotspot analysis was confined to the rectangular bounding grid used for vetting the VPS positions (Fig 4). Position data was divided into three timeframes: 1) dredge operational in lower area, 2) dredge operational in upper area, and 3) no dredge operations occurring.

Gaps in a fish's swim patch occur because its position is only determined when the tag sends a ping and pings are not always detected on enough receivers in a VPS for a position to be determined. The longer the time between positions will likely lead to more error when describing swim behavior because we are assuming a linear swim path between two positions. To reduce the effects of describing swim behavior only fish tracks extending at least 1 km with no consecutive points greater than 15 minutes apart were used for the swim behavior analyses. Each path was classified according to whether the direction of movement was upstream or downstream, as well as whether the dredge was at the downstream site, the upstream site or

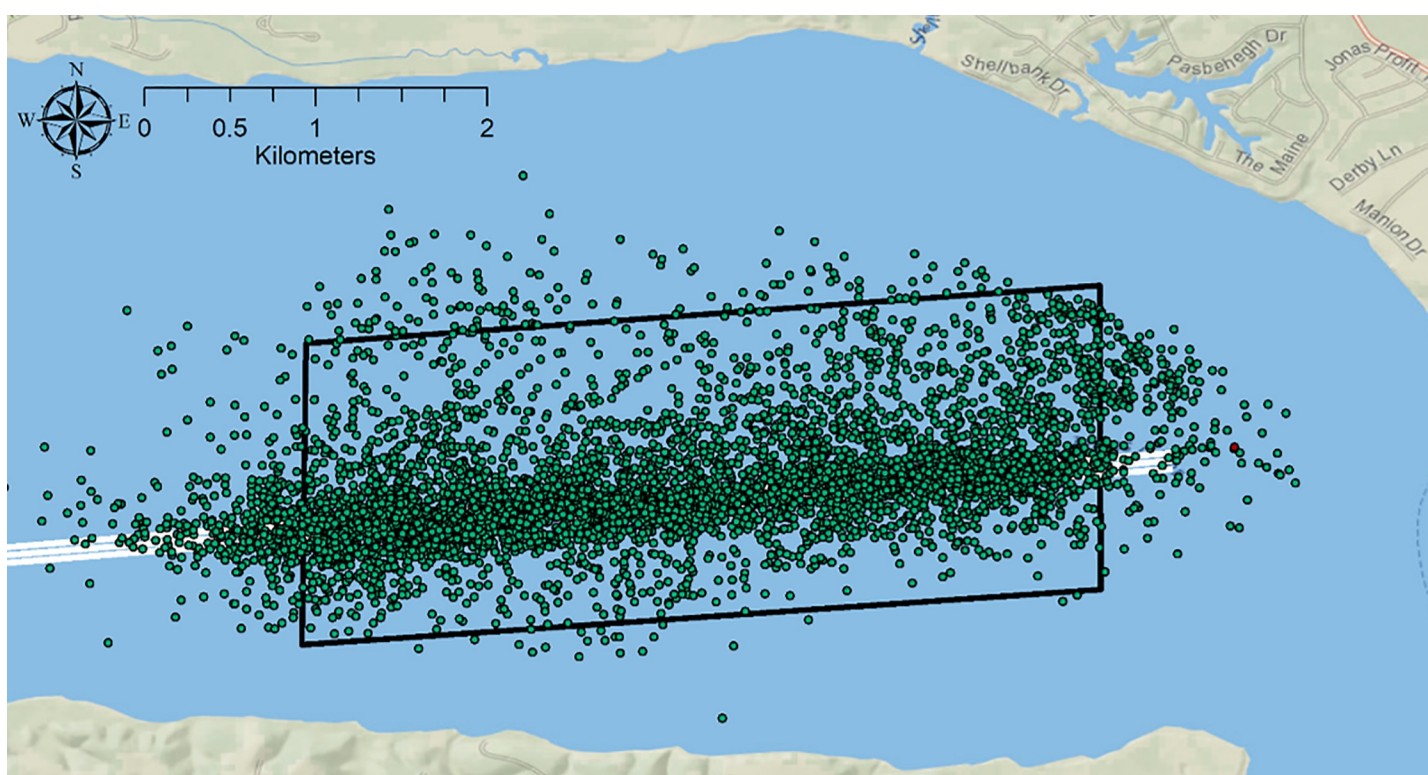

**Fig 4. Points showing all adult Atlantic sturgeon positions generated by the Vemco Position System array.** The black box is the extent of the Getis Gi* Hotspot analysis. In total 99 adults were represented and 5839 positions were recorded, of which 5,011 were used after the points were vetted for accuracy. Basemap reprinted from Ocean Basemap under a CC BY license, with permission from ESRI, original copyright 2012.

absent. Swim speeds were determined by measuring the distance between all points of an individual track and dividing by elapsed time. Permutation tests with 25,000 iterations using the 'independence_test' function in R Statistical Software, version 3.5.2, package "coin 1.3–0" were run comparing ATS swim speed versus movement direction through the study area and whether the dredge was working in the lower area, upper area, or if the dredge was absent [15]. Another aspect of swim behavior analyzed was how much an ATS deviated from a straight swam path, *i.e.* meandered, while moving through the study area. The difference in the actual distance swam and the straight-line distance between the first and last point of the track was calculated in meters. Using the same methods as the speed permutation test, separate permutation tests were run on how much ATS meandered while traversing through the study area.

## Water quality

Water temperature is a critical factor influencing the timing of ATS migration to spawning habitat [16]. Hourly water temperature measurements were obtained 2 km downstream of the study site by The National Oceanic and Atmospheric Administration. [17]. Average hourly water temperature measurements were estimated from July 20 to November 3, 2017. In previous years, male fall spawning ATS tended to start migrating upstream when the daily mean water temperatures fell below 29°C (Matthew Balazik, Virginia Commonwealth University, Unpublished Data).

## Dredge

The hydraulic-cutterhead dredge, *Lexington*, conducted maintenance dredging during the study period. The *Lexington* is 61 m long, 125 m wide, 3 m draft barge with a 2000 HP pump

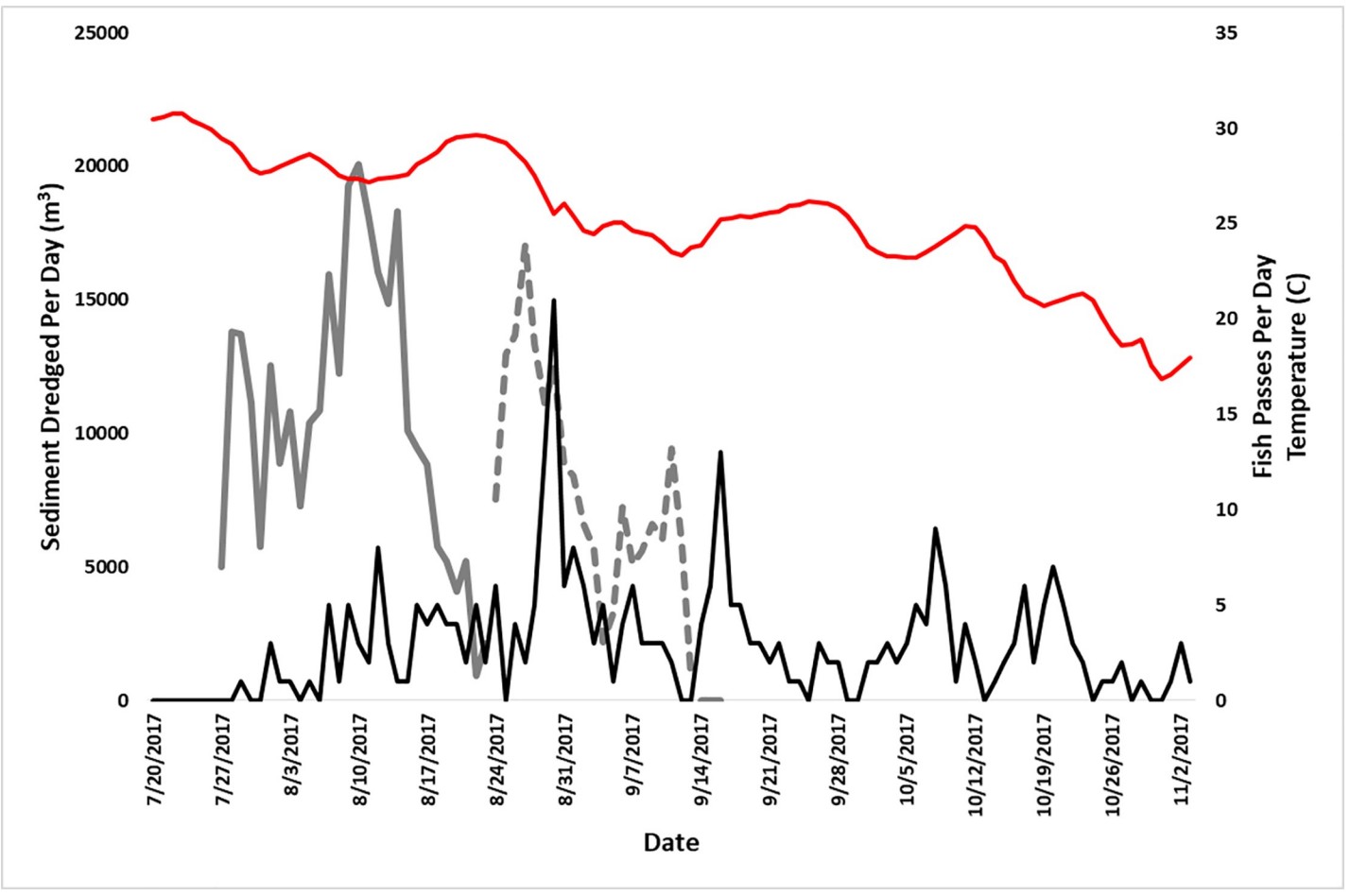

**Fig 5. The primary Y-axis shows how much material was removed from the lower (solid gray line) and upper (dashed gray line) dredge area.** The secondary Y-axis shows temperature (red line) and number of adult Atlantic sturgeon moving passed the dredge or through the area (black line) each day. Serendipitously dredging started when water temperatures dropped to levels that trigger upstream spawning migrations.

connected to a 51 cm diameter intake pipe located just shipward of the cutterhead. The cutterhead tip had a 1.4 m diameter. Dredge operations started at 1500 on July 27 and where continuous until the project ended at 1700 on September 16, 2017. Dredging occurred in the lower area from July 27 to August 23 and September 12 to 14. Dredging occurred in the upper area from August 23 to September 12 and September 15 and 16 (Fig 5). The dredge removed 278,056 m³ of sediment from the lower dredge area and 227,311 m³ from the upper area (Fig 5). The dredge's location was determined at 30 s intervals using the United States Army Core of Engineers Automatic Identification System Analysis Package (AISAP) tool. All dredged material was placed via a partially floating and sinking pipeline to a pre-determined site located south of the navigation channel (Fig 2).

## Results and discussion

During the study period, 106 adult ATS (7 female and 99 male) were detected in the study area, 103 of which swam past the active dredge. Accurate positions could not be determined for 7 males, but the dates of movement are known. A total of 5,839 fine-scale positions were determined by the VPS of which 5,011 from 7 females and 92 males were used for analysis

(Fig 4). All ATS were detected by the supplemental array. The 7 males that had no fine-scale positions estimated moved upstream in late August during the peak of upstream migration (Fig 5). In the lower area, 48 different adult ATS made a total of 88 passes by the active dredge (Fig 5). Seventy-four adults moved past the dredge a total of 129 times while the dredge was working in the upper area (Fig 5). After dredging finished on September 16, 93 adults traversed the study area 125 times (Fig 5).

Three male ATS moved through and staged upstream of the study area prior to dredge operations. As in previous years, most adults started moving upstream when daily mean water temperatures at the water quality buoy fell below 29˚C (Fig 5). Mean daily water temperatures fell below 29˚C in the study area on July 30. ATS staging downstream started migrating upstream to spawning habitat on July 29 when the dredge was working in the lower area. Adult activity gradually increased in the study area but then decreased as water temperatures began increasing around August 16 (Fig 5). As water temperatures dropped relatively quickly starting August 28, upstream migration activity peaked while the dredge was working in the upper area (Fig 5). Based on real-time receivers around hypothesized spawning habitat, the peak of the 2017 spawning occurred from September 5 to 14. When dredging operations were finishing in the upper area there was a spike in migrating adults on September 15 as the post-spawn ATS began to return to the ocean. Most telemetered male ATS stayed around spawning habitat for weeks after the peak spawning season but started migrating downstream in early October (Fig 5).

The dredge did not limit ATS movement or cause mortalities within the James River. All ATS that reached the study area while the dredge was working moved upstream to spawning habitat. Telemetry data showed that ATS swam past dredge operations 217 times. Over half (n = 59, 56%) of the spawning ATS made one pass by the dredge. Forty-four adults (41%) passed the active dredge multiple times with one ATS passing the dredge 13 times (Fig 6). If the active dredge and associated noise were a deterrent it is unlikely adult ATS would make unnecessary, repeat trips past the dredge. Three males moved upstream in June before dredge operations and downstream in October, completely missing the dredge activity altogether (Fig 6).

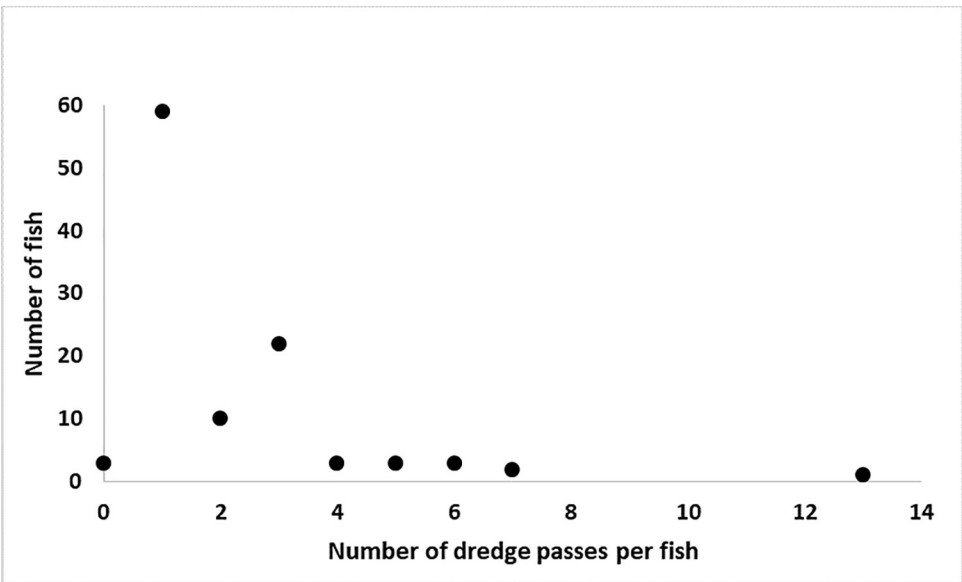

**Fig 6. The number of times adult Atlantic sturgeon moved past the hydraulic-cutterhead dredge.** The Y-axis is the total number of fish that passed the dredge the number of times listed on the X-axis.

Hot Spot Analysis showed adult positions were more densely packed in the immediate channel area during upstream and downstream migration (Fig 7). When the dredge was working in the lower and upper areas, most of the adults were making upstream migrations to spawning habitat. Adult positions were concentrated in the channel during upstream migration regardless of the presence or absence of an operating dredge in the immediate area and avoided areas away from the channel (Fig 7). During the main downstream migration after dredging was complete, adult ATS still had high densities in the channel but were more scattered throughout the study area compared to when ATS were mostly moving upstream to spawning habitat (Fig 7).

It is possible that dredge activity attracts migrating adult ATS to the channel. However, it's more likely the adult fish naturally utilize the channel during upstream migration due to depth, current flow or some other factors. Catch and telemetry data show migrating adults have a high affinity for deep channels [18] in the James River and results from this study suggest the preference continues even if dredging is occurring. Based on passive telemetry data during the previous 5 years, telemetered adult ATS have traversed active dredge operations hundreds of times in the James River and data indicate that no mortalities of telemetered ATS resulted. The VPS telemetry data from this study indicate that no telemetered adults were killed during dredging operations and none stopped upstream progress due to the dredge.

ATS traversed the VPS study area 298 times, 170 of which satisfied our criteria for individual movement analysis (Fig 8). ATS swim speed or meandering did not vary significantly when factoring for dredge location and ATS swim direction (Table 1, Fig 9). These data

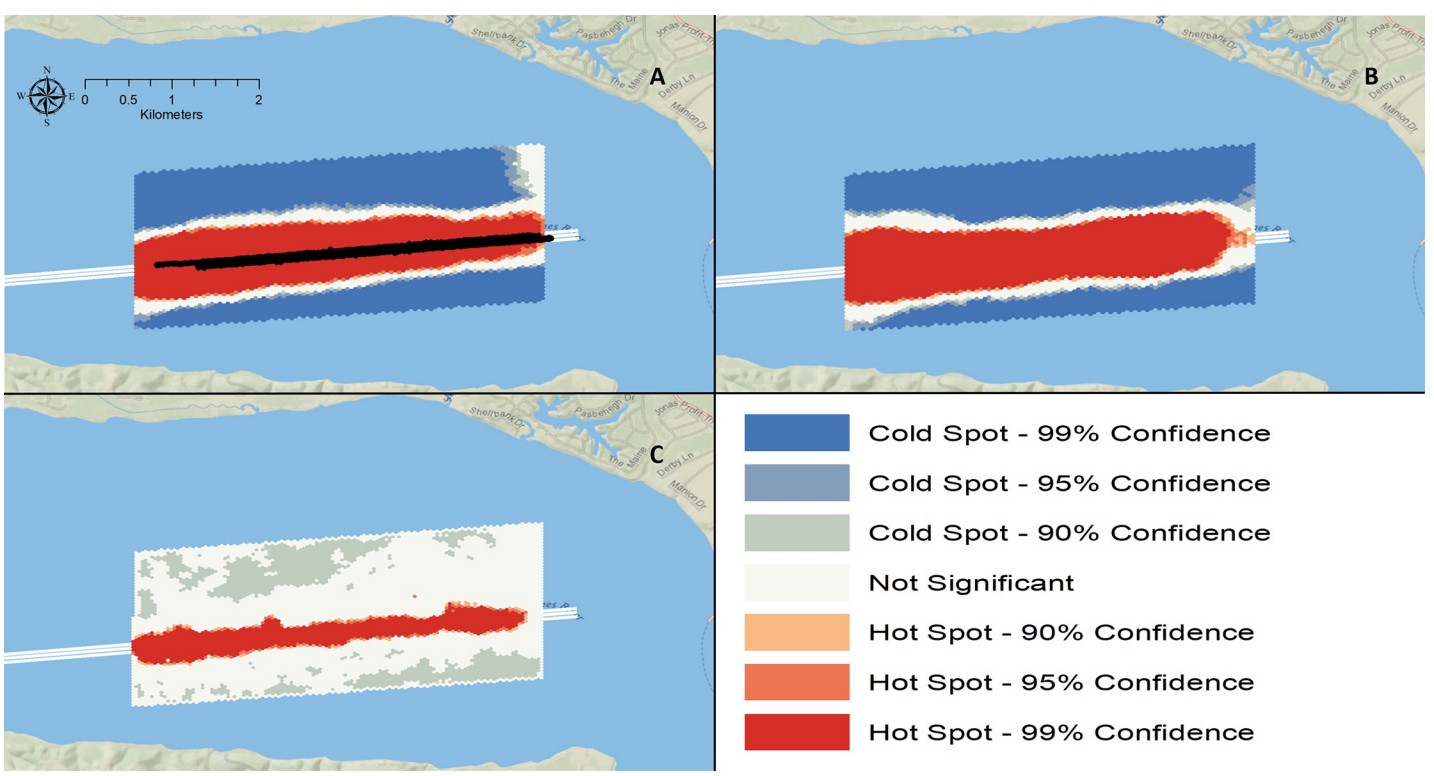

**Fig 7.** Results of the Getis Gi* Hotspot Analysis for when the dredge was working in the lower area (Panel A), working in the upper area (Panel B) and when no dredging was occurring (Panel C). The black dots in the middle of the hot spot in Panel A shows where the dredge was working during the lower area. The white polygon marks the federal navigation channel. All three plots show Atlantic sturgeon's preference for the navigation channel either with or without a hydraulic-cutterhead dredge conducting dredge operations. Basemap reprinted from Ocean Basemap under a CC BY license, with permission from ESRI, original copyright 2012.

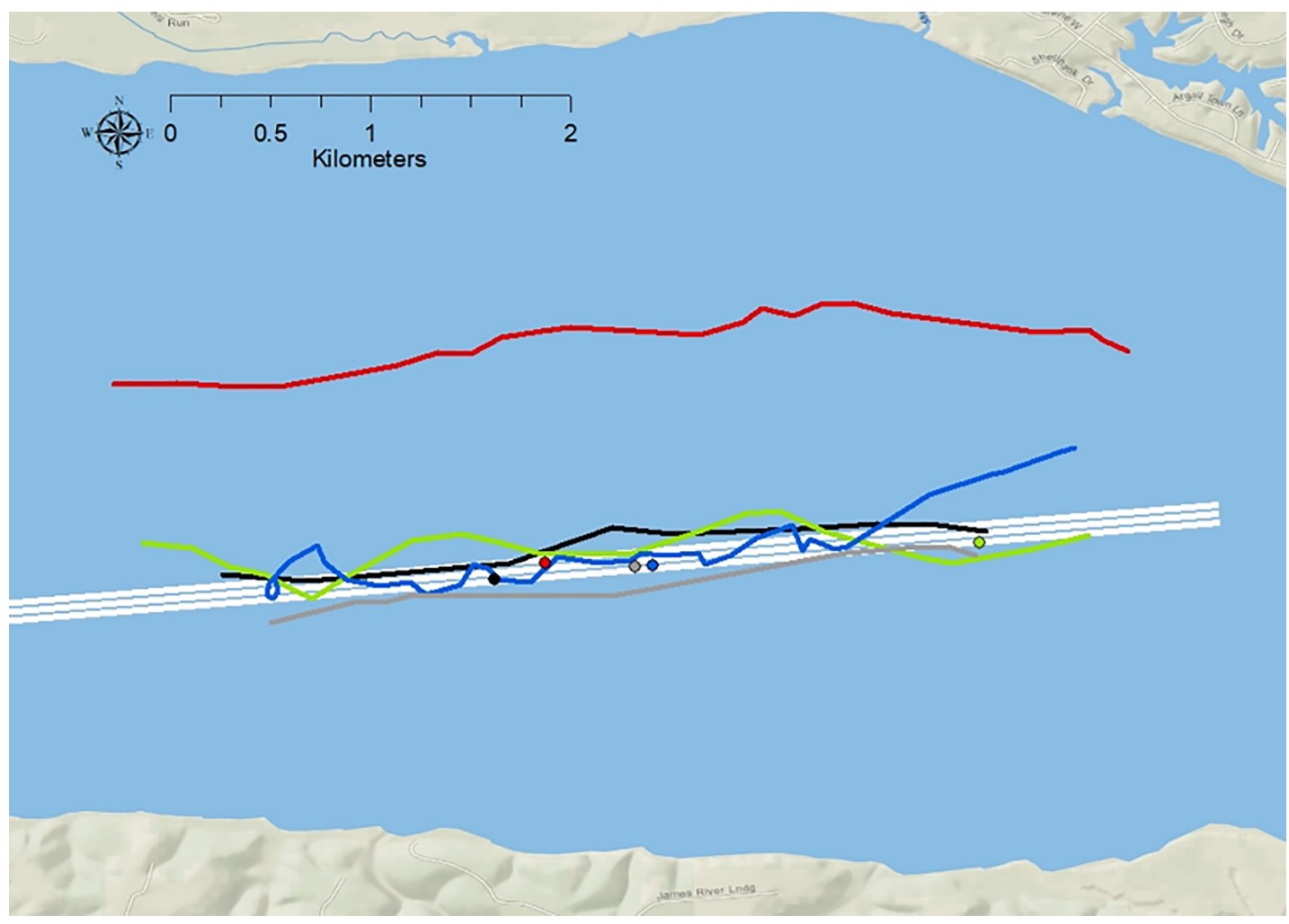

**Fig 8. Individual tracks (colored lines) of adult Atlantic sturgeon swimming in an upstream to spawning habitat during dredge operations.** The colored dots represent the dredge location for when the sturgeon of the same color line moved through the area. Basemap reprinted from Ocean Basemap under a CC BY license, with permission from ESRI, original copyright 2012.

suggest that there was no difference in swim speed or meandering in the VPS area whether the dredge was operational within the lower or upper areas or not operational.

## Conclusions

The results of this study indicate that hydraulic-cutterhead dredging in the study area does not deter adult ATS from migrating to spawning habitat and had no observable effect on swim behavior. This study along with Reine et al. [18] found no evidence of any subadult or adult ATS being killed by a hydraulic-cutterhead dredge in the James River. The combined evidence

**Table 1. Results of permutation tests comparing swim speed and meandering behavior factoring in swim direction and dredge location.**

| Fish movement direction | Number of tracks | | | Average swim speed (m/s) | | | | Average meandering (m) | | | |
|---|---|---|---|---|---|---|---|---|---|---|---|
| | Absent | Lower | Upper | Absent | Lower | Upper | p-value | Absent | Lower | Upper | p-value |
| Upstream | 8 | 34 | 36 | 0.71 | 0.83 | 0.78 | 0.43 | 458 | 454 | 297 | 0.44 |
| Downstream | 57 | 15 | 20 | 0.57 | 0.59 | 0.67 | 0.76 | 472 | 511 | 447 | 0.96 |

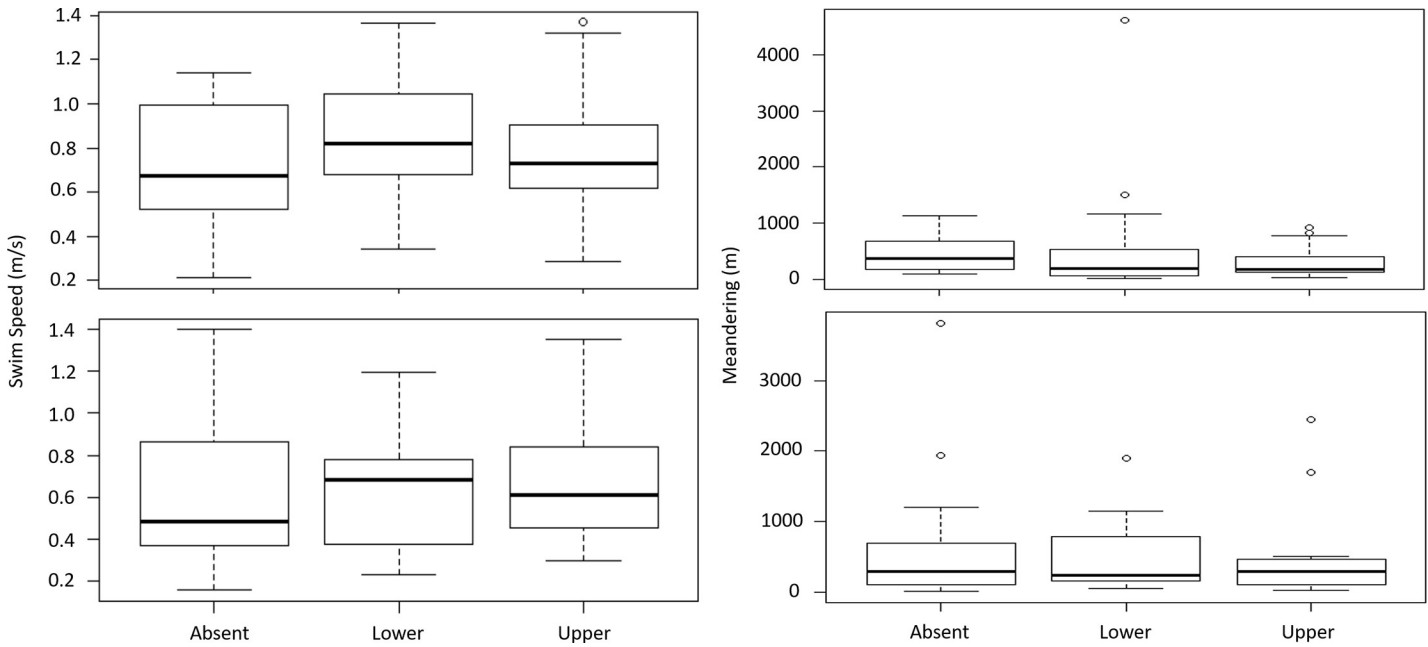

**Fig 9. Left boxplots show swim speed (meters per second) and right boxplots show meandering (difference in actual swim path from a linear swim path) when the dredge was not in the area (Absent), when the dredge was working in the lower area (Lower) and when the dredge was working in the upper area (Upper).** The top boxplots are when Atlantic sturgeon where swimming upstream through the study area while the bottom boxplots are when Atlantic sturgeon were swimming downstream through the study area. Permutation tests showed no significant difference between swim speed and dredge location for upstream movements (p = 0.429) and dredge location for downstream movements (p = 0.756). Permutation tests also showed no significant difference between meandering and dredge location for upstream movements (p = 0.44) and for dredge location and downstream movements (p = 0.96).

of this study and past data suggests that the physical presence of a hydraulic-cutterhead dredge and associated underwater sounds during the spawning season does not hinder ATS spawning migration.

Data from this study suggest that dredge restrictions in this area where eggs and larvae are likely not to be present may not be an effective management policy. This study was conducted in a relatively broad, >4 km wide, section of the James River; however, ATS spawning occurs further upstream where the river is typically <200 m wide. While in this study many adults passed within 100 m of the dredge, there may be a different reaction in a much narrower channel. Mimicking this type study in a narrower channel or closer to spawning habitat would provide useful information for resource managers. More work is also needed to look at potential impacts of various dredge types on juvenile ATS in different environments. It would also be beneficial to conduct fish movement-dredge VPS studies to examine other species of concern, such as striped bass (*Morone saxatilis*) and Alosines, which are protected by the anadromous fish environmental dredge window.

## Supporting information

**S1 Data.**
(ZIP)

## Acknowledgments

We would like to thank Albert Spells (US Fish and Wildlife Service), Eric Hilton, Patrick McGrath and Matthew Fisher (Virginia Institute of Marine Science), Robert Greenlee

(Virginia Department of Game and Inland Fisheries), Martin Balazik, Thiwaporn Balazik, George Trice, Kelly Place, and Charles Frederickson (field assistants), Douglas Clarke (Clarke Environmental), the USACE Norfolk District, and the crew of Dredge Lexington for assisting with this project. This manuscript represents VCU Rice Rivers Center publication 91. The authors declare that they have no conflict of interest.

## Author Contributions

**Conceptualization:** Matthew Balazik.

**Formal analysis:** Matthew Balazik, Michael Barber.

**Funding acquisition:** Matthew Balazik, Greg Garman.

**Investigation:** Matthew Balazik, Michael Barber.

**Methodology:** Matthew Balazik, Michael Barber, Safra Altman.

**Project administration:** Matthew Balazik.

**Resources:** Matthew Balazik.

**Visualization:** Michael Barber.

**Writing – original draft:** Matthew Balazik, Kevin Reine, Alan Katzenmeyer, Aaron Bunch, Greg Garman.

**Writing – review & editing:** Matthew Balazik, Safra Altman, Kevin Reine, Alan Katzenmeyer, Aaron Bunch, Greg Garman.

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
