## [Decision Letter · Decision Letter 0]

6 Jan 2020

PONE-D-19-29177

Dredging activity and associated sound have negligible effects on adult Atlantic sturgeon migration to spawning habitat in a large coastal river

PLOS ONE

Dear Dr. Balazik,

Thank you for submitting your manuscript to PLOS ONE. After careful consideration, we feel that it has merit but does not fully meet PLOS ONE’s publication criteria as it currently stands. Therefore, we invite you to submit a revised version of the manuscript that addresses the points raised during the review process.

Your manuscript has been reviewed by two reviewers. Although the external reviewers express interest in the general subject area of the paper, they also express a series of reservations that preclude publication of the paper in PLoS ONE in its current form. However, if you feel that you can suitably address the concerns and issues raised by the reviewers, I would be willing to consider a revised manuscript. Also, please be advised that the revised manuscript may be subject to re-evaluation.

We would appreciate receiving your revised manuscript by Feb 20 2020 11:59PM. To enhance the reproducibility of your results, we recommend that if applicable you deposit your laboratory protocols in protocols.io, where a protocol can be assigned its own identifier (DOI) such that it can be cited independently in the future. For instructions see: http://journals.plos.org/plosone/s/submission-guidelines#loc-laboratory-protocols

We look forward to receiving your revised manuscript.

Kind regards,

Zuogang Peng, Ph.D.

Academic Editor

PLOS ONE

Journal Requirements:

2. In your Methods section, please provide additional location information of the study area, including geographic coordinates for the data set if available.

6. Your ethics statement must appear in the Methods section of your manuscript. If your ethics statement is written in any section besides the Methods, please move it to the Methods section and delete it from any other section. Please also ensure that your ethics statement is included in your manuscript, as the ethics section of your online submission will not be published alongside your manuscript.

7. We note that Figures 1, 3, 7 and 8 in your submission contain map images which may be copyrighted. All PLOS content is published under the Creative Commons Attribution License (CC BY 4.0), which means that the manuscript, images, and Supporting Information files will be freely available online, and any third party is permitted to access, download, copy, distribute, and use these materials in any way, even commercially, with proper attribution. For these reasons, we cannot publish previously copyrighted maps or satellite images created using proprietary data, such as Google software (Google Maps, Street View, and Earth). For more information, see our copyright guidelines: http://journals.plos.org/plosone/s/licenses-and-copyright.

You may seek permission from the original copyright holder of Figures 1, 3, 7 and 8 to publish the content specifically under the CC BY 4.0 license. 

If you are unable to obtain permission from the original copyright holder to publish these figures under the CC BY 4.0 license or if the copyright holder’s requirements are incompatible with the CC BY 4.0 license, please either i) remove the figure or ii) supply a replacement figure that complies with the CC BY 4.0 license. Please check copyright information on all replacement figures and update the figure caption with source information. If applicable, please specify in the figure caption text when a figure is similar but not identical to the original image and is therefore for illustrative purposes only.

8.  Thank you for stating the following in your Competing Interests section: 

'No'

9. Please amend either the title on the online submission form (via Edit Submission) or the title in the manuscript so that they are identical.

Reviewers' comments:

Reviewer's Responses to Questions

**Comments to the Author**

1. Is the manuscript technically sound, and do the data support the conclusions?

Reviewer #1: Yes

Reviewer #2: No

2. Has the statistical analysis been performed appropriately and rigorously? 

Reviewer #1: Yes

Reviewer #2: No

3. Have the authors made all data underlying the findings in their manuscript fully available?

Reviewer #1: Yes

Reviewer #2: Yes

4. Is the manuscript presented in an intelligible fashion and written in standard English?

Reviewer #1: Yes

Reviewer #2: No

5. Review Comments to the Author

Reviewer #1: Overall this is a very well written manuscript and the authors are commended for their efforts.

The study sample size is extremely limited regarding females. Suggesting dredging does not have an impact on adult ATS migration is a rather sweeping claim and a bit presumptuous given the study’s samples and geographic size of the study area. I strongly encourage the authors limit this study to male subjects or address the limited number of females in the study within the discussion.

In figure 1, it is unclear what the colors along the dredge lines indicate (“the colored dots within the channel are where the hydraulic-cutterhead dredge was working during the study period”). Do warmer colors signify the amount of time a dredge working in an area or the time of year the dredge was working? Please clarify.

To this point, why did the authors set up the VPS at the downstream site over the upstream site? Please incorporate the reasoning into the manuscript.

The authors report the number of fish in the study area used in the hot spot analysis but was there a minimum number of positions per fish required? Did these positions need to be captured within a certain window of time, similar to the swimming speed requirements? Incorporating a fish's positions in July versus September might be very different. Was a seasonal component or individual effect included? The authors elude to the differences in behaviors across seasons so this should be accounted for in the analyses.

Figure 7 is very interesting and I believe the authors should expand text in the discussion around this figure. Its very interesting that in plot C the fish are in a very narrow cluster along the channel (presumed preferred habitat) but in plots A and B the cluster is much broader and fish are avoiding banks. To me, this suggests dredging does influence migratory behavior. From your analyses it does not appear to impact the upstream or downstream movement or meandering in the lower dredging area but there appears to be an impact that should be discussed.

Remove or expand figure 8. Either include all fish's paths or none.

In general, the manuscript's discussion could be expanded and incorporate related studies.

Reviewer #2: The study assesses the effects of dredging on migration of sturgeon for purportedly spawning purposes. It is laudable in the sense that the study uses existing fish with transmitters and essentially, employs an array of Vemco receivers. However, my main two concerns are the writing styles and the interpretations or inferences from their study.

I found this manuscript to be very difficult to read and follow given the number of inverted, double negative and redundant sentences. I have made annotations on a pdf of your manuscript to highlight some problem areas and suggested some potential edits. I think you would be able to relay your message much more effectively if you conducted a very hard edit on your manuscript.

Your study was about the affects of dredging on migrating sturgeon, I recommend you adhere to that in your conclusions. You cannot make any statement about spawning success or efficacy of policy on recruitment because you didn't test it in your study. I have provided comments about the structure of your manuscript. For example, your conclusion is not a conclusion. You shouldn't be presenting new material in the conclusion. I think your abstract captures your study nicely, your conclusion doesn't.

Minor comments

Can you conduct statistical analyses to support your observations on initiation of movement (e.g., logistic regression or GLM using temperature and upstream movement)? It could strengthen your study.

This study was conducted predominately on males. I understand that you are relying on previously transmittered fish and therefore have to rely on movement, but could you address potential limitations in your discussion (e.g., sex or size effect). Additionally, could you provide metrics of the fish in the study with the caveat that they were of that size upon sample.

Several figures are, in my opinion, not necessary (e.g., Figures 2 and 4); Figure 6 would be better if it was a histogram.

Do not start sentences with acronyms.

6. PLOS authors have the option to publish the peer review history of their article (what does this mean?). If published, this will include your full peer review and any attached files.

Reviewer #1: No

Reviewer #2: No

---

## [Author Response · Author response to Decision Letter 0]

25 Jan 2020

2. In your Methods section, please provide additional location information of the study area, including geographic coordinates for the data set if available.

 A figure (Figure 1) was added to provide additional location information. The lat/lon points for the fish are also in the supplemental information.

 Added a line saying this was public area so no special permits were required to access the study site. 

6. Your ethics statement must appear in the Methods section of your manuscript. If your ethics statement is written in any section besides the Methods, please move it to the Methods section and delete it from any other section. Please also ensure that your ethics statement is included in your manuscript, as the ethics section of your online submission will not be published alongside your manuscript.

Added Ethics Statement at the beginning of the methods section.

7. We note that Figures 1, 3, 7 and 8 in your submission contain map images which may be copyrighted. All PLOS content is published under the Creative Commons Attribution License (CC BY 4.0), which means that the manuscript, images, and Supporting Information files will be freely available online, and any third party is permitted to access, download, copy, distribute, and use these materials in any way, even commercially, with proper attribution. For these reasons, we cannot publish previously copyrighted maps or satellite images created using proprietary data, such as Google software (Google Maps, Street View, and Earth). For more information, see our copyright guidelines: http://journals.plos.org/plosone/s/licenses-and-copyright.

1. You may seek permission from the original copyright holder of Figures 1, 3, 7 and 8 to publish the content specifically under the CC BY 4.0 license. 

8. Thank you for stating the following in your Competing Interests section: 

'No'

9. Please amend either the title on the online submission form (via Edit Submission) or the title in the manuscript so that they are identical.

 I’m sorry but we can not tell a difference between what is on the online submission form and the title on the manuscript.

Reviewers' comments:

Reviewer's Responses to Questions

Comments to the Author

1. Is the manuscript technically sound, and do the data support the conclusions?

Reviewer #1: Yes

Reviewer #2: No

2. Has the statistical analysis been performed appropriately and rigorously?

Reviewer #1: Yes

Reviewer #2: No

3. Have the authors made all data underlying the findings in their manuscript fully available?

Reviewer #1: Yes

Reviewer #2: Yes

4. Is the manuscript presented in an intelligible fashion and written in standard English?

Reviewer #1: Yes

Reviewer #2: No

5. Review Comments to the Author

Reviewer #1: Overall this is a very well written manuscript and the authors are commended for their efforts.

The study sample size is extremely limited regarding females. Suggesting dredging does not have an impact on adult ATS migration is a rather sweeping claim and a bit presumptuous given the study’s samples and geographic size of the study area. I strongly encourage the authors limit this study to male subjects or address the limited number of females in the study within the discussion.

We understand that the sample size is highly skewed towards males over females. However, this does not change the fact that all the females in the study moved passed the dredge and reached spawning habitat. Even with the small female sample size, the claim is still valid that the dredge did not have any noticeable effects on adult sturgeon moving up to spawning habitat. We added text in the discussion to point out there was a relatively small amount of females but the point is still valid. 

In figure 1, it is unclear what the colors along the dredge lines indicate (“the colored dots within the channel are where the hydraulic-cutterhead dredge was working during the study period”). Do warmer colors signify the amount of time a dredge working in an area or the time of year the dredge was working? Please clarify.

The different colors show where the dredge was working each day. We added text to the figure description. The figure would have been too crowded to show what color each day represents; however, the description of when the dredge was working on each day is described later in the methods section.

To this point, why did the authors set up the VPS at the downstream site over the upstream site? Please incorporate the reasoning into the manuscript.

We did not have enough receivers to cover both areas. We chose the narrower river width so our receivers would cover most of the overall width of the river. We added text to the methods section.

The authors report the number of fish in the study area used in the hot spot analysis but was there a minimum number of positions per fish required? Did these positions need to be captured within a certain window of time, similar to the swimming speed requirements?

No, all the points that fit the vetting criteria described in the position analysis were used to the hot spot analysis. This is stated in the methods section. If the only points used for the hot spot analysis had to pass the same conditions as the swim speed analysis there would not have been enough data to the analysis to show hot and cold spots.

Incorporating a fish's positions in July versus September might be very different. Was a seasonal component or individual effect included? The authors elude to the differences in behaviors across seasons so this should be accounted for in the analyses.

We agree that there might be a seasonal component in regards to pre/post spawn fish. The fish are moving back and forth during from July-November and it is possible that pre and post spawn fish move differently. During the July/early September most the fish were making their initial run to spawning grounds while in late September/November the fish were leaving for the year. Since dredging ended in the middle of September, comparing figures 7a and 7b to figure 7c is like taking into account pre and post spawning movements. 

Figure 7 is very interesting and I believe the authors should expand text in the discussion around this figure. Its very interesting that in plot C the fish are in a very narrow cluster along the channel (presumed preferred habitat) but in plots A and B the cluster is much broader and fish are avoiding banks. To me, this suggests dredging does influence migratory behavior. From your analyses it does not appear to impact the upstream or downstream movement or meandering in the lower dredging area but there appears to be an impact that should be discussed.

This is a common misinterpretation of the figure, we thought it as well but had ERSI (maker of the tool) explain things to us. So according to the analysis in general the fish were closer to the channel, and therefore the dredge, and avoided the fringes (7a and b) compared to when the dredge had stopped working (7c). The fish were more dispersed when the dredge was not working which means the dredge had some sort of attraction effect. This is highly unlikely and is already noted in the discussion section. The only way to prove this would be to have the dredge work during October/November and not during August/September while still having the same water quality parameters. The dredge can not wait so late in the season to work because shipping would have to stop. 

Remove or expand figure 8. Either include all fish's paths or none.

If all paths are plotted (170) it would be just on big blur because all the lines would be overlapping and wouldn’t make sense. We think it helps provide a scale to how close the fish swam passed the active dredge. 

In general, the manuscript's discussion could be expanded and incorporate related studies. 

The discussion was expanded; however, the only truly related study that tracks telemetered fish around an active dredge is already cited in this paper. 

Reviewer #2: The study assesses the effects of dredging on migration of sturgeon for purportedly spawning purposes. It is laudable in the sense that the study uses existing fish with transmitters and essentially, employs an array of Vemco receivers. However, my main two concerns are the writing styles and the interpretations or inferences from their study.

I found this manuscript to be very difficult to read and follow given the number of inverted, double negative and redundant sentences. I have made annotations on a pdf of your manuscript to highlight some problem areas and suggested some potential edits. I think you would be able to relay your message much more effectively if you conducted a very hard edit on your manuscript.

We conducted a very hard edit to the manuscript. We might have a different writing style from reviewer 2. We would like to point out the reviewer 1 said the manuscript was very well written.

Your study was about the affects of dredging on migrating sturgeon, I recommend you adhere to that in your conclusions. You cannot make any statement about spawning success or efficacy of policy on recruitment because you didn't test it in your study. I have provided comments about the structure of your manuscript. For example, your conclusion is not a conclusion. You shouldn't be presenting new material in the conclusion. I think your abstract captures your study nicely, your conclusion doesn't.

The text was modified to stating that adults reached spawning habitat. We addressed the comments in the PDF.

Minor comments

Can you conduct statistical analyses to support your observations on initiation of movement (e.g., logistic regression or GLM using temperature and upstream movement)? It could strengthen your study.

We would have to look at multiple years of data and beyond the scope of this study. Physical cues to sturgeon movements are going to be a major point for another paper we plan to write. 

This study was conducted predominately on males. I understand that you are relying on previously transmittered fish and therefore have to rely on movement, but could you address potential limitations in your discussion (e.g., sex or size effect). Additionally, could you provide metrics of the fish in the study with the caveat that they were of that size upon sample.

This was addressed in comments by reviewer 1. These fish were proven to be adults during initial capture. Most we tagged over 4 years prior to this study and growth rate is highly variable due to the fact that some years adults skip spawning and likely grow more than fish that do spawn and whether the fish spend their ocean time north or south of VA. We think there are too many variables to account for a size effect during this study when there is so much variability of growth from fish tagged 4+ years ago. 

Several figures are, in my opinion, not necessary (e.g., Figures 2 and 4); Figure 6 would be better if it was a histogram.

Figure 4 will be removed. However, the setup shown in figure 2 is something that researchers are always asking for us to explain. We feel keeping figure 2 will help other conduct their own VPS studies.

Do not start sentences with acronyms.

---

## [Decision Letter · Decision Letter 1]

20 Feb 2020

Dredging activity and associated sound have negligible effects on adult Atlantic sturgeon migration to spawning habitat in a large coastal river

PONE-D-19-29177R1

Dear Dr. Balazik,

We are pleased to inform you that your manuscript has been judged scientifically suitable for publication and will be formally accepted for publication once it complies with all outstanding technical requirements.

With kind regards,

Zuogang Peng, Ph.D.

Academic Editor

PLOS ONE

Additional Editor Comments (optional):

Reviewers' comments:

Reviewer's Responses to Questions

**Comments to the Author**

1. If the authors have adequately addressed your comments raised in a previous round of review and you feel that this manuscript is now acceptable for publication, you may indicate that here to bypass the “Comments to the Author” section, enter your conflict of interest statement in the “Confidential to Editor” section, and submit your "Accept" recommendation.

Reviewer #2: All comments have been addressed

2. Is the manuscript technically sound, and do the data support the conclusions?

Reviewer #2: Yes

3. Has the statistical analysis been performed appropriately and rigorously? 

Reviewer #2: N/A

4. Have the authors made all data underlying the findings in their manuscript fully available?

Reviewer #2: Yes

5. Is the manuscript presented in an intelligible fashion and written in standard English?

Reviewer #2: Yes

6. Review Comments to the Author

Reviewer #2: The authors did an excellent job addressing most of my concerns and edits. Personally, I think the paper is stronger. I still feel the manuscript is qualitative vs quantitative and some of their conclusions could be strengthened by statistical analyses.

7. PLOS authors have the option to publish the peer review history of their article (what does this mean?). If published, this will include your full peer review and any attached files.

Reviewer #2: Yes: Tim Haxton

---

## [Editor Report · Acceptance letter]

24 Feb 2020

PONE-D-19-29177R1 

Dredging activity and associated sound have negligible effects on adult Atlantic sturgeon migration to spawning habitat in a large coastal river 

Dear Dr. Balazik:

I am pleased to inform you that your manuscript has been deemed suitable for publication in PLOS ONE. Congratulations! Your manuscript is now with our production department. 

With kind regards,

on behalf of

Dr. Zuogang Peng 

Academic Editor

PLOS ONE